# Contextual and neural representations of sequentially complex animal vocalizations

## Abstract

Holistically exploring the perceptual and neural representations underlying animal communication has traditionally been very difficult because of the complexity of the underlying signal. We present here a novel set of techniques to project entire communicative repertoires into low dimensional spaces that can be systematically sampled from, exploring the relationship between perceptual representations, neural representations, and the latent representational spaces learned by machine learning algorithms. We showcase this method in one ongoing experiment studying sequential and temporal maintenance of context in songbird neural and perceptual representations of syllables. We further discuss how studying the neural mechanisms underlying the maintenance of the long-range information content present in birdsong can inform and be informed by machine sequence modeling.

## 1   Introduction

Systems neuroscience has a long history of decomposing the features of complex signals under the assumption that they can be untangled and explored systematically, part-by-part. For example in audition, much of the early work in exploring physiological representations of signals involves playing back sine tones, white noise, or other single-variable-modulated acoustic signals. While these approaches have led to a number of advances in understanding circuits and systems underlying auditory cognition, many neural phenomena cannot be understood without exploring systems in more biologically realistic environments.

In many cases, introducing biological realism into controlled experiments requires uncovering the complex feature spaces underlying signals. For example, the neuroscience of human language is based on a rich understanding of the phonological, semantic, and syntactic features of speech and language. In contrast, the communicative spaces of many model organisms in auditory neuroscience are more poorly understood, leading to a very small number of model organisms having the necessary tools for study. In birdsong, for example, biophysical models of song production that have been developed for zebra finches do not capture the dynamics of the dual-syrinx vocal tract of European starlings. More species general approaches to modeling communication would increase the accessibility of more diverse and more systematic explorations of animal communication systems in neuroscience.

Here, we propose a method based upon recent advances in generative modeling to explore and exploit the vocal and communicative spaces of animals more generally. We show that unsupervised generative and dimensionality reduction machine learning models can be used to learn a latent representation of various animal communicative systems, requiring few prior assumptions about the animal's communication. We can then sample from these latent spaces to systematically explore neural and perceptual representations of biologically relevant acoustic spaces with complex features. We show that this method is successful in species as diverse as songbirds, primates, insects, cetaceans, and amphibians, and in recording conditions both in the lab and in the field. We demonstrate this

technique in one ongoing experimental paradigm exploring the effects of sequential context on neural representations of syllables in songbirds. The method we outline here is currently under development, however, a recent version is available as a resource on GitHub[1].

## 2  Latent representations of animal communication

Dimensionality reduction and generative models uncover latent structure and nonlinear features in complex audio and visual data [1]. In recent years, these techniques have played an important role in uncovering motifs in behavioral data [2-4] and circuit computations in neural data [5-7].

The methods used to learn latent feature representations are various and have different utility based upon use-case. For example, topologically motivated embedding techniques like UMAP and TSNE are often used for learning data motifs (Fig. 1) but are less useful for generating smooth or novel data manifolds. Generative models like Generative Adversarial Networks (GANs) and Variational Autoencoders (VAEs) are useful for interpolating between and smoothly generating signals, but learn a predefined latent distribution (e.g. uniform or Gaussian) that holds little information about the underlying data. Architectural constraints also result in different representations; for example, convolutional versus recurrent neural networks respect different aspects of data. Because use cases differ when exploring animal communication, we implement and explore a number of these models on audio, such that we produce a series of latent representations that can be used for latent modeling of animal communication in a similar manner as in neural and ethological mapping, as well as stimuli generation for physiological and behavioral probing.

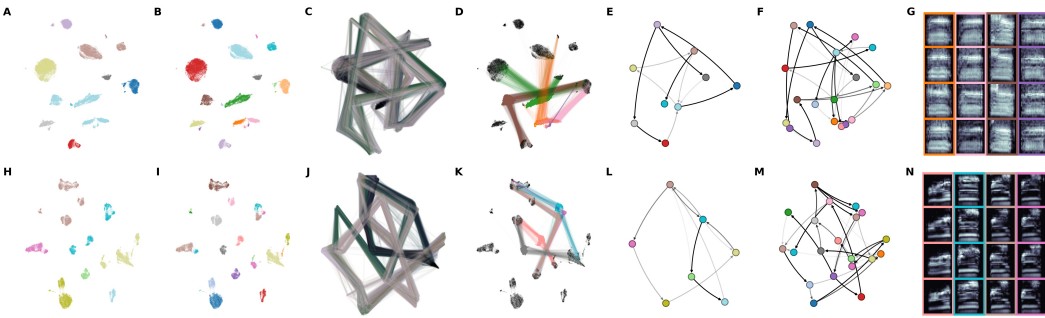

Figure 1: Latent comparisons of hand- and algorithmically-clustered Bengalese finch song. A-H are from a dataset produced by Nicholson et al., [9] and H-N are from a dataset produced by Koumura et al., [10] (A,H) UMAP projections of syllables of Bengalese finch song, colored by a combination of hand labels and supervised labels. (B,I) Algorithmic labels. (C, J) Transitions between syllables, where color represents time. (D,K) Comparing the transitions in one hand-labelled category vs multiple algorithmic labels. (E,L) Markov model from hand labels. (F,M) Markov model from clustered labels. (G,H) Examples of syllables from multiple algorithmic clusters falling under a single hand-labelled cluster.

## 3  Neural and behavioral representations of context

A major question in both machine learning and computational cognitive neuroscience is how sequential information can be maintained in order to best predict future events. In machine learning, recent advances such as gated recurrent neural networks, transformer models, and autoregressive models have resulted in progressively improved modelling of sequences, however the flexibility with which sequence models can capture long-range relationships are largely constrained (e.g. [10]) and require vastly different training regimes than the human brain. At the same time, the active maintenance of information in the human brain, through recurrent feedback loops between prefrontal cortex and basal ganglia, show many similarities with sequence models in machine learning [11]. Likewise, songbird basal ganglia and frontal cortex analogous structures actively maintain temporal information, and songbird temporal structure exhibits long-range temporal dependencies that parallel those seen in language [12].

---

[1]Link omitted for anonymity until publication

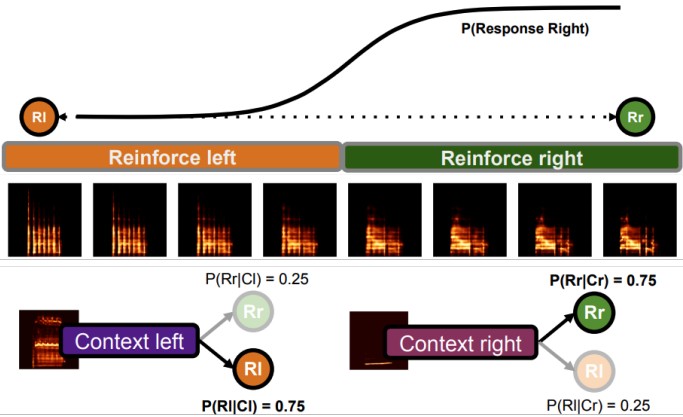

Figure 2: Outline of the context-dependent perception task. Birds are tasked with classifying a smooth morph between syllables generated from a VAE, generating a psychometric function of classification behavior. Sequential-contextual cues that precede the classified syllables are given to bias the psychometric function.

In the present experiment we explore how sequential context is maintained in the songbird brain. To this end, we train a songbird to classify regions of a VAE-generated latent space of song, and manipulate the perception of those regions of space based upon sequential-contextual information (Fig 2). Specifically, we interpolate between syllables of European starling song projected into latent space. We train a starling to classify the left and right halves of the interpolation using an operant-conditioning apparatus (Fig. 4). We then provide a contextual syllable preceding the classified syllable that holds predictive information over the classified syllable (Fig 2 bottom). We hypothesize that the perception of the boundary between the classified stimuli shifts as a function of the context cue. We model this hypothesis using Bayes rule:

$$\underbrace{P(x_{true} \mid x_{sensed}, cue)}_{\text{posterior}} \propto \underbrace{P(x_{sensed} \mid x_{true}, cue)}_{\text{likelihood}} \underbrace{P(x_{true} \mid cue)}_{\text{prior}}$$

When a stimulus varies upon a single dimension $x$ (the interpolation), the perceived value of $x$ is a function of the true value of $x$ and contextual information (Fig. 3 left). The initial behavioral results of our experiment confirmed our hypotheses (Fig. 3). We additionally performed acute extracellular neural recordings from two auditory regions in songbird brain (NCM, CMM) during stimulus playback under anesthesia. We found single neuron responses to stimuli vary continuously as a function of interpolation point.

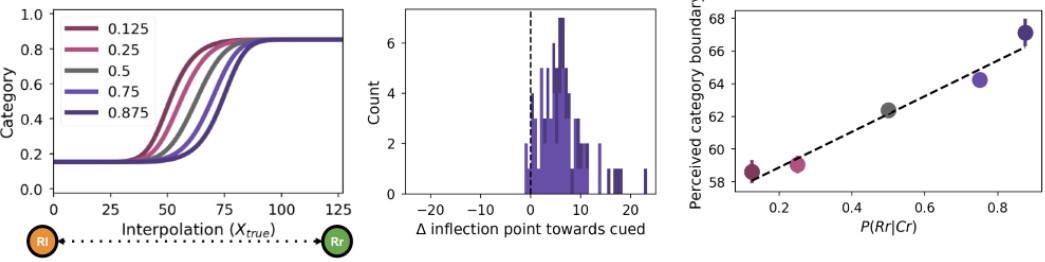

Figure 3: Results for behavior. (left) The Bayesian model predicts that the inflection point of the psychometric classification function will be shifted as a function of the context cue (colored lines). (center) We found that in nearly all behavioral conditions, the inflection point was shifted in the direction of the cue. (right) We found a significant correlation between the cue probability, and the fit categorical boundary using a 4-parameter logistic function (r(223) = 0.60, p < 0.001).

Because this experiment requires active behavior and maintenance of information, we are currently implementing a chronic version of this experiment, where neural populations are recorded during

active behavior. For this chronic version of the experiment, we designed a custom Raspberry-Pi-based behavioral apparatus that interfaces with Open Ephys for extracellular acute physiological recordings (Fig. 4).

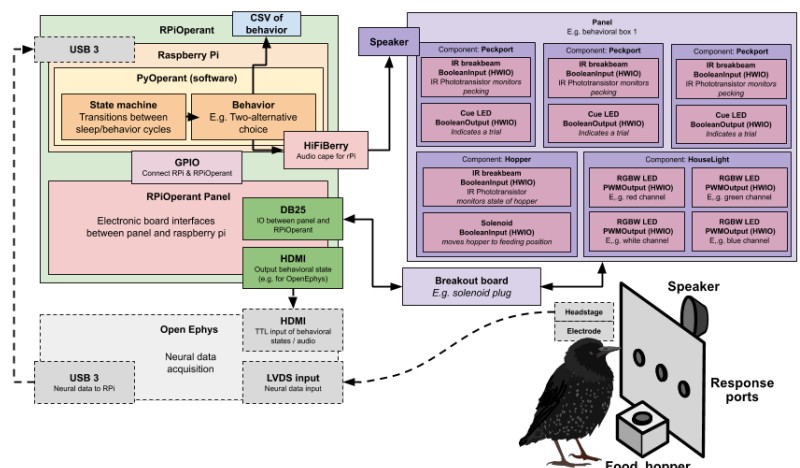

Figure 4: Rig designed for the current study. Behaviors are controlled through a custom interface based upon the Raspberry Pi. Neural data is recorded through an Open Ephys board.

Future work will scale the current behavior up to the more complex sequences produced by birds in a naturalistic setting. Using this paradigm, we hope to directly explore the computational circuits involved in the complex long-range patterns that emerge in songbird communication. We expect that understanding how the long-range hierarchical organization of songbird communication is maintained and represented will be informative in generating new methodologies for maintaining long-range information content in machine learning and AI. Likewise, we expect that improvements in sequence modeling in machine learning will provide testable hypotheses for mechanisms underlying neural circuits of context representation.

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
