# OpenReview forum: "Contextual and neural representations of sequentially complex animal vocalizations"
_NeurIPS.cc/2019/Workshop/Neuro_AI — Submitted to Real Neurons & Hidden Units @ NeurIPS 2019_

### Official Review · AnonReviewer1 · 2019-09-23
**Useful direction but weakly informative results**

**Clarity:** 2

**Comment:**

While the general idea of using generative latent variable methods to systematically explore behavioral space is compelling, the authors provide little evidence for most of their key claims. For example, in the introduction they say they will show that the method works across species and conditions, but then give only a birdsong example. Similarly, they claim single neuron responses vary continuously with interpolation point but show no data to support this. Overall, while the overarching idea is interesting, results feel very preliminary and weakly presented.


**Category:**

Not applicable

**Clarity Comment:**

The figures were confusing (e.g. it is not clear exactly what is depicted in Fig 1, nor what its main point is) and it was rather hard to follow the key results. It also, for example, wasn’t clear whether the context was a binary or real-valued signal.


**Evaluation:**

2: Poor

**Importance:**

3: Important

**Importance Comment:**

The authors address an important problem of developing a general, species-independent approach to quantifying animal vocalizations. Their approach is to use generative dimensionality reduction techniques to learn low-dimensional representations of vocalizations and use this to systematically interpolate between vocalization patterns to map out their perceptual organization in brains. This is a compelling idea, but the presented results shed little light on it.


**Intersection:**

2: Low

**Intersection Comment:**

There was little intersection between neuro and AI, besides using simply using machine learning algorithms to classify/generate behavioral signals.


**Rigor Comment:**

While the authors make some attempt to survey properties of a few dimensionality reduction techniques in Fig 1, this is not very clear, and the authors don’t make a systematic attempt to compare dimensionality reduction techniques in the context of their novel approach, nor to identify key parameters or constraints for successful operation.


**Technical Rigor:**

2: Marginally convincing

---

### Official Review · AnonReviewer2 · 2019-09-24
**Potentially interesting but more rigorous results are needed**

**Clarity:** 2

**Comment:**

The idea of this paper is potentially interesting but the results are primitive and need more rigorous analyses.

**Category:**

AI->Neuro

**Clarity Comment:**

The figure 1 is confusing and hard to follow, it seems like A-G and H-N are two different examples from two datasets but in paper, A-H are from a dataset and H-N are from the other dataset. Figure 4 is filled with a lot of blocks but explained with very few words.

**Evaluation:**

2: Poor

**Importance:**

3: Important

**Importance Comment:**

The authors first propose to utilize generative modelling and dimension reduction technique to get a general low-dimensional representation of animal vocal spaces and then by sampling from the latent space, they systematically explore neural and perceptual representations of biologically relevant acoustic spaces with complex features. The direction is interesting but current results are primitive and not enough.

**Intersection:**

2: Low

**Intersection Comment:**

The paper doesn't really relate to real neurons but mainly focus on utilizing artificial neural network techniques on animal behavior.

**Rigor Comment:**

Although the authors claim they implement and explore a number of models to produce a series of latent representations, only the result of VAE is reported and there is no explanation why VAE is preferred to others. The author also claim their method is successful in different species but only songbirds result is provided.

**Technical Rigor:**

2: Marginally convincing

---

### Official Review · AnonReviewer3 · 2019-09-27
**probing the perceptual space of bird song generation informed by generative models**

**Clarity:** 2

**Comment:**

The project is innovative and promising. At the current stage, the results seem to be rather preliminary, but the project might still be a good candidate for the workshop because the research design is innovative and potentially disruptive, so it might spark a good discussion.

**Category:**

AI->Neuro

**Clarity Comment:**

The scientific questions are clearly explained, the methodology clear, but the results seem rather vague and preliminary.


**Evaluation:**

3: Good

**Importance:**

3: Important

**Importance Comment:**

This work addresses how VAE could help to model and characterize bird song generation. The authors propose to use a VAE to learn a low-dimensional representation of bird songs. Interpolations in the low-dimensional representations between stimuli are then used in classifications tasks to probe the perception of boundaries between stimuli and (in the future) the corresponding neural representations. The paper seems to be an innovative research agenda rather than a finalized project.

**Intersection:**

3: Medium

**Intersection Comment:**

The paper uses machine learning generally and VAE specifically mostly as a fitting toolbox, not as a model of the neural circuit. While the project might be innovative on the neuroscience community, especially for sequence generation in animal vocalization, the relevance of the project for the machine learning community might be rather limited.

**Rigor Comment:**

The proposed model class is rather standard in the machine learning community but seems novel for the specific task of animal vocalization. While the results are rather preliminary, the proposed experimental of inferring low-dimensional representations of bird song and using the for behavioral experiments in combination with neural recordings seems innovative. Machine learning generally and VAE specifically seems mostly to be used as a fitting tool, not as a model of the neural circuit.

**Technical Rigor:**

3: Convincing

---

### Decision · Program_Chairs · 2019-10-01

**Decision:**

Reject

**Comment:**

Unfortunately, we had more submissions than we could accept and based on the review process, we have decided not to accept your submission.  Nevertheless, thank you for your submission and interest in our workshop.